# Impact of Low-Dose Methotrexate–Adalimumab Combination Therapy on the Antibody Response Induced by the mRNA-1273 SARS-CoV-2 Vaccine: Case of an Elderly Patient with Rheumatoid Arthritis

**DOI:** 10.3390/vaccines9080883

**Published:** 2021-08-09

**Authors:** Yves Michiels, Nadhira Houhou-Fidouh, Gilles Collin, Jérôme Berger, Evelyne Kohli

**Affiliations:** 1Community Pharmacy, Center for Primary Care and Public Health (Unisanté) Lausanne 1011, University of Lausanne, 1015 Lausanne, Switzerland; jerome.berger@unisante.ch; 2Pharmacie Michiels, 21600 Longvic, France; 3Laboratoire de Virologie, Hôpital Bichat, Sorbonne Paris Cité, Université Paris Diderot, AP-HP, 75018 Paris, France; nadira.houhou@aphp.fr (N.H.-F.); gilles.collin@aphp.fr (G.C.); 4School of Pharmaceutical Sciences, University of Geneva, 1205 Geneva, Switzerland; 5Institute of Pharmaceutical Sciences of Western Switzerland, University of Lausanne, 1015 Lausanne, Switzerland; 6UMR INSERM/uB/AGROSUP 1231, Team 3 HSP-Pathies, Labellisée Ligue Nationale Contre le Cancer and Laboratoire d’Excellence LipSTIC, 21000 Dijon, France; evelyne.kohli@u-bourgogne.fr; 7UFR des Sciences de Santé, Université de Bourgogne, 21000 Dijon, France; 8CHU, 2100 Dijon, France

**Keywords:** SARS-CoV-2, mRNA1273 vaccine, antibody response, methotrexate, adalimumab

## Abstract

Patients with rheumatoid arthritis (RA) are treated with drugs that may impact their immune responses to SARS-CoV-2 vaccines. We describe here the anti-Spike (anti-S) IgG and neutralizing antibody responses induced by the mRNA-1273 SARS-CoV-2 vaccine in a 78-years-old patient with RA, who received a low-dose combination therapy of methotrexate and adalimumab, shortly before vaccine administration. Both near-normal and impaired immune responses to vaccines have been reported previously in patients treated with these drugs. Our case report shows that, even at low doses, combined methotrexate-adalimumab therapy can be associated with a weak immune response to the mRNA1273 vaccine in elderly patients.

## 1. Introduction

Patients with rheumatoid arthritis (RA) are treated with drugs that affect their immune responses. Data published, so far, have indicated that patients treated with conventional disease modifying antirheumatic drugs (DMARDs), alone or in combination with biotherapies, are at no higher risk of hospitalization or mortality than the general population, except for those receiving glucocorticoids at dosages >10 mg/day [1,2,3,4,5,6]. RA patients are currently advised to continue their antirheumatic treatment, follow protective and social distancing measures, and to be vaccinated as soon as possible. However, data on the antibody responses, induced by the various SARS-CoV-2 vaccines, in such patients are still limited. In particular, there are no data concerning immune responses in older patients on methotrexate-anti-TNF-α combination therapy. Methotrexate is known to alter T lymphocyte proliferation, and anti-TNF-α agents, such as adalimumab, have been reported to weaken the IgG humoral response and to cause a reduction in CD27 memory B cell levels. Anti-TNF-α agents have already been shown to impact the antibody response to the hepatitis A and B vaccines, and methotrexate has been found to impair the response to both the 13- and 23-valent pneumococcal vaccines [7,8]. Indeed, a temporary 2-week interruption of methotrexate therapy has been proposed, as a measure to improve the immunogenicity of the influenza vaccine [9]. On the other hand, near-normal vaccine immune responses have been reported in some studies [7,8,10]. RNA vaccines represent an innovative strategy, applied to respond to the global SARS-CoV-2 outbreak; however, there is a lack of available data on the antibody response, induced by mRNA SARS-CoV-2 vaccines, in patients of different age groups receiving various immunosuppressive therapies. In this case report, we describe the anti-S IgG and neutralizing antibody responses, induced by the mRNA-1273 vaccine (manufactured by Moderna), in a 78-year-old patient with RA being treated with a low-dose combination of methotrexate and the anti-TNF-α biotherapy, adalimumab (patient 1). The vaccine response in a control patient, who was aged 73 years and was not receiving immunosuppressant therapy, was also assessed (patient 2).

## 2. Case Description

Patient 1: A 78-year-old woman with RA, without comorbidities, was being treated with methotrexate (7.5 mg per week), folic acid (5 mg per week), and adalimumab 40 mg (one injection per month). The patient was also receiving calcium 500 mg (one tablet per day), cholecalciferol 400 IU (one tablet per day), cholecalciferol 100,000 IU (one ampoule per month), and lansoprazole 30 mg (one tablet per day).

The patient was vaccinated with the mRNA-1273 vaccine, according to the recommended protocol for first injection, on 18 January 2021. This was followed by a second dose on 15 February 2021 (Table 1). The patient received adalimumab 4 days before the first dose and 1 day before the second dose of the vaccine. Methotrexate was administered on the day of vaccination for both vaccine doses.

A quantitative serologic test (LIAISON^®^ SARS-CoV-2 TrimericS IgG assay; DiaSorin, Saluggia, Italy), to determine the levels of IgG directed against the trimeric form of the SARS-CoV-2 spike protein, was performed on a blood sample taken on 25 February 2021, i.e., 38 days (D-38) after the first vaccine injection The test showed that anti-S IgG antibodies were present, at a level of 193 AU (Arbitrary Units)/mL (501.8 BAU (Binding Antibody Units)/mL) (Table 2), indicating the presence of a moderate positive response to the first dose of the vaccine [7]. Two further tests were then conducted on the same blood sample: another anti-S IgG test (Architect^®^ anti-spike test; Abbott, Rungis, France) and a pseudo-neutralization test (Iflash Orgentec™, Trappes, France) to determine the level of neutralizing antibodies [11].

The results revealed an anti-S IgG titer of 1366 AU/mL (threshold: 50 AU/mL) and a neutralizing antibody titer of 106.46 AU/mL (threshold: 10 AU/mL) (Table 2).

A second round of serologic tests was performed on a further blood sample, taken from the patient on 15 March 2021, i.e., 53 days after the first injection (D-53). The results showed that the level of anti-S IgG at D-53 was very similar to that observed at D-38, according to both methods used: the DiaSorin LIAISON^®^ SARS-CoV-2 TrimericS assay (IgG: 184 AU/mL; 478.4 BAU/mL) and the Abbott Architect^®^ anti-spike test (1391 AU/mL). The neutralizing antibody level remained in the same range (177.60 AU/mL) as that measured at D-38 (Table 2).

Patient 2 (control): The control patient was a 73-year-old woman who was not receiving any immunosuppressant drugs but was being treated with levothyroxine 125 mg (one tablet per day), nebivolol 5 mg (one tablet per day), irbesartan 300 mg (one tablet per day), hydrochlorothiazide 25 mg (one tablet per day), acetylsalicylic acid 160 mg (one tablet per day), cholecalciferol 100,000 IU (one dose every 3 months), ezetimibe 10 mg (one tablet per day), and simvastatin 40 mg (one tablet per day). The first dose of the mRNA-1273 SARS-CoV-2 vaccine was administered on 18 March 2021, followed by a second dose on 15 April 2021. The results of the serological tests, to determine the level of anti-S IgG in samples, from this control patient are shown in Table 2. The neutralizing antibody titers in this control patient were 10 times higher than those in patient 1.

## 3. Discussion

Vaccination against SARS-CoV-2 is recommended for patients with RA who are being treated with immunosuppressants [4]. However, only partial data are available on the antibody responses induced by the various SARS-CoV-2 vaccines in these patients and, in particular, there are no data concerning vaccine responses in older patients receiving methotrexate-anti-TNF-α combination therapy.

Both near-normal and impaired immune responses to vaccines have been reported previously in patients treated with these drugs [7,8,10]. In the case of mRNA SARS-CoV-2 vaccines, the preprint report of Deepak et al. revealed that anti-S IgG titers were 2- to 3-fold lower in patients on methotrexate, compared to those in untreated controls, but no significant differences in neutralizing antibody titers were observed; whereas patients receiving anti-TNF-α agents had lower levels of neutralizing antibodies, but not lower anti-S IgG titers compared to immunocompetent controls [14]. In another study, Geisen et al. reported that vaccination with an mRNA SARS-CoV-2 vaccine resulted in lower levels of anti-S and neutralizing antibodies in patients with RA that were treated with an anti-TNF-α agent and/or leflunomide compared to control subjects [15]. Furthermore, two recent studies have also found weaker anti-S IgG or weaker anti-receptor binding domain total immunoglobulin responses to SARS-CoV-2 vaccines in patients with Crohn’s disease that were treated with anti-TNF-α agents, compared to those on vedolizumab therapy or control patients, respectively. Neutralizing antibody levels were not analyzed in these two studies [16,17]. The levels of anti-S IgG, and of neutralizing antibodies, in our patient were low and were at least 4 times and 10 times lower, respectively, than those in the control patient. Moreover, anti-S IgG levels in patient 1 appeared to have reached a plateau after the first injection of the mRNA-1273 vaccine, suggesting that the treatment, and/or the timing of administration (very close to that of the vaccine), had an impact on the memory response to vaccination. Altogether, this report shows that, even at low doses, the combined methotrexate-adalimumab therapy was associated with a weak immune response to the mRNA1273 vaccine in an elderly patient.

Although this case study lacks the statistical power needed to draw any conclusions, it highlights the difficulties associated with determining the level and duration of vaccine-induced protection in older patients being treated with immunosuppressants. In order to better support patients with RA, and to provide appropriate guidance for these patients on the prevention of COVID-19 infection, more data are needed on the immunogenicity of the various SARS-CoV-2 vaccines, from a larger population of patients receiving various immunosuppressive treatments and at different dosages. In particular, further studies are essential with a view to easing social distancing measures and to provide guidance on the question of an additional vaccine dose (which has already been proposed in some situations), as well as on the question of the timing of administration of immunosuppressants, both before and after vaccination.

## Figures and Tables

**Table 1 vaccines-09-00883-t001:** Dates of administration of the vaccine and of the methotrexate–adalimumab combination therapy.

	Administration	Date
	Adalimumab (40 mg)	14 January 2021
	Methotrexate (7.5 mg)	18 January 2021
First-dose vaccination	Injection mRNA-1273 SARS-CoV-2	18 January 2021
	Adalimumab (40 mg)	14 February 2021
	Methotrexate (7.5 mg)	15 February 2021
Second-dose vaccination	Injection mRNA-1273 SARS-CoV-2	15 February 2021

**Table 2 vaccines-09-00883-t002:** Antibody responses after the first (D-38) and the second (D-53) dose of mRNA-1273 SARS-CoV-2 vaccine.

	Test	Antibody Titer after Administration of the mRNA-1273 SARS-CoV-2 Vaccine	
		D-38	D-53
Patient 1	DiaSorin LIAISON^®^ SARS-CoV-2 TrimericS IgG™ assay [12]	193 AU/mL (501.8 BAU/mL)	184 AU/mL (478.4 BAU/mL)
	Abbott Architect™ anti-spike [13]	1366 AU/mL	1391 AU/mL
	Iflash Orgentec™ neutralization test [11]	106.46 AU/mL	177.60 AU/mL
Patient 2	Diasorin LIAISON^®^ SARS-CoV-2 TrimericS IgG™ assay [12]	>800 AU/mL (>2080.0 BAU/mL)	>800 AU/mL (>2080.0 BAU/mL)
	Iflash Orgentec™ neutralization test [11]	2870 UA/mL	2710 UA/mL

## Data Availability

Data available on request due to restrictions, e.g., privacy or ethic.

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
