# Peer review of "Impact of Low-Dose Methotrexate–Adalimumab Combination Therapy on the Antibody Response Induced by the mRNA-1273 SARS-CoV-2 Vaccine: Case of an Elderly Patient with Rheumatoid Arthritis"

_vaccines, 2021, doi:10.3390/vaccines9080883_

Round 1

Reviewer 1 Report

In this manuscript, the authors provide case report on the effect of combined immune-suppressive therapy of methotrexate and adalimumab on the immune response to the mRNA 1273 vaccine in 78-year old patient with rheumatoid arthritis. They observed that both the anti-S IgG and neutralizing antibody levels were lower in this patient compared to a control patient not on immune-suppressive therapy.  This interesting observation need to be further evaluated in a larger patient population to determine the effect of different therapies, dosages and timeline of the treatments on the immune response to mRNA1273 vaccine.   

This case report is well written and explores the topic of great interest. I would recommend the publication of this report in its present form with a few spelling and grammar corrections. 

Author Response

We thank the reviewer for his interest in our case.

- We have slightly modified the end of the discussion as follows: “Although this case study is

not statistically sufficient to draw any conclusions, it highlights the difficulties….”

- As suggested, we have reread the manuscript and hope to have improved English

Reviewer 2 Report

This paper addresses the very interesting question of whether or not additional considerations should be given to rheumatoid arthritis patients who take both methotrexate and adalimumab when considering the SARS-CoV-2 vaccination. The authors present a very preliminary analysis that provides no concrete answers to this question, but the importance of asking this question is justified. I would hope that the authors will take this research to the next logical stage and provide empirical evidence that can shed light on this important issue.

Author Response

We thank the reviewer for his interest in our case.

- We have now slightly modified the end of the discussion as follows: “Although this case study

is not statistically sufficient to draw any conclusions, it highlights the difficulties….”

Reviewer 3 Report

The results of this clinical case report reveal what we have also observed in our clinical practice. The publication of case reports like this one is useful in order to optimize therapy in patients with rheumatic diseases who must undergo anti-SARS CoV-2 vaccination in order to obtain an adequate immune response to the vaccine.

Author Response

(The authors gave the same response as above.)
